

# Evidence for the adaptive parsing of non-communicative eye movements during joint attention interactions

Ayeh Alhasan and  Nathan Caruana

[1] School of Psychological Sciences, Macquarie University, Sydney, New South Wales, Australia
[2] Perception in Action Research Centre, Macquarie University, Sydney, New South Wales, Australia

## ABSTRACT

During social interactions, the ability to detect and respond to gaze-based joint attention bids often involves the evaluation of non-communicative eye movements. However, very little is known about how much humans are able to track and parse spatial information from these non-communicative eye movements over time, and the extent to which this influences joint attention outcomes. This was investigated in the current study using an interactive computer-based joint attention game. Using a fully within-subjects design, we specifically examined whether participants were quicker to respond to communicative joint attention bids that followed predictive, as opposed to random or no, non-communicative gaze behaviour. Our results suggest that in complex, dynamic tasks, people adaptively *use* and *dismiss* non-communicative gaze information depending on whether it informs the locus of an upcoming joint attention bid. We also went further to examine the extent to which this ability to track dynamic spatial information was specific to processing gaze information. This was achieved by comparing performance to a closely matched non-social task where eye gaze cues were replaced with dynamic arrow stimuli. Whilst we found that people are also able to track and use dynamic non-social information from arrows, there was clear evidence for a relative advantage for tracking gaze cues during social interactions. The implications of these findings for social neuroscience and autism research are discussed.

## INTRODUCTION

Joint attention is the ability to intentionally coordinate our attention with a social partner so that we are both attending to the same object or event at the same time (*Bruner, 1974*; *Carpenter & Liebal, 2011*; *Hobson, 2005*; *Tomasello, 1995*). The early development of joint attention is pivotal in supporting the later development of language (*Akhtar, Dunham & Dunham, 1991*; *Charman, 2003*; *Dawson et al., 2004*) and social-cognitive skills, including the ability to understand and predict the mental states and perspectives of others (*Mundy, 2003*; *Mundy, 2016*; *Mundy, 2018*; *Mundy & Jarrold, 2010*; *Mundy & Neal, 2000*; *Mundy & Newell, 2007*).

Corresponding authors
Ayeh Alhasan,
ayeh.alhasan@mq.edu.au
Nathan Caruana,
nathan.caruana@mq.edu.au

Until recently, most experimental studies of 'joint attention' required participants to observe and respond to a single gaze cue on each trial, often in a non-interactive context (see *Caruana et al., 2017b*; *Frischen, Bayliss & Tipper, 2007*; *Nation & Penny, 2008*; *Dalmaso, Castelli & Galfano, 2020*; *McKay et al., 2021* for reviews). However, in genuine daily interactions gaze cues are observed in the context of a *dynamic* and experiential flow of perception and action (*Heft, 2003*). The dynamic nature of gaze requires us to parse constant streams of eye movements to determine when a gaze shift is communicative and intentionally signals an opportunity for joint attention. According to the Relevance Theory and the 'ostensive-inferential communication' model, this differentiation is achieved by identifying the relevance of ostensive signals (*e.g.*, eye contact) and using them to infer the communicative intent of a social partner (*Sperber & Wilson, 1986*; *Wilson & Sperber, 2002*). This notion of relevance is not limited to the transmission of communicative intentions. Rather, it is a more general property of information processing that can be applied to other categories of information, including non-communicative but informative gestures. In the context of gaze-based joint attention, it is not yet clear whether and to what extent people evaluate the relevance of non-communicative eye movements in an interactive context. As such, the current study aimed to examine whether, and to what extent, humans parse and are influenced by the non-communicative eye movements that precede eye contact and subsequent bids for joint attention. This endeavour is critical for fully understanding how gaze signals are used to guide social coordination in realistic dynamic interactions.

## Ostensive eye contact

Ostensive signals such as eye contact are argued to signal communicative intent, which can increase the perceived relevance and communicativeness of subsequent behaviours (including eye movements; *Sperber & Wilson, 1986*; *Wilson & Sperber, 2002*). The ability of eye contact to capture attention is observed in infants within their first year, and is believed to play a crucial role in signaling relevant social information during interactions (*Behne, Carpenter & Tomasello, 2005*; *Csibra, 2010*; *Wilson & Sperber, 2002*). Indeed, studies show that eye contact, especially during live interactions, induces psychophysiological arousal that results in increased alertness and attention (*Gale et al., 1975*; *Helminen, Kaasinen & Hietanen, 2011*; *Hietanen, Peltola & Hietanen, 2020*; *Kleinke & Pohlen, 1971*; *Nichols & Champness, 1971*). Other studies and related models (*e.g.*, fast-track modulator model; *Senju & Johnson, 2009*), also indicate that observing eye contact results in a rapid and automatic activation of the social brain network and subsequently upregulates social-cognitive processing—including the representation of others' mental states—*via* a subcortical pathway (*Conty et al., 2007*; *Mares et al., 2016*; *Senju & Johnson, 2009*). However, it is not yet clear if and how these eye contact signals are used to identify communicative joint attention bids when embedded in a dynamic sequence of eye movements.

## Non-communicative eye movements

In very recent work, we conducted the first experimental interrogation of the extent to which humans process and are influenced by non-communicative eye movements leading up to a joint attention opportunity (*Caruana et al., 2020*; *Caruana et al., 2017a*). This revealed that

humans do sensitively parse and differentiate communicative and non-communicative eye movements during dynamic gaze-based interactions. Specifically, in our most recent study (*Caruana et al., 2020*), we designed two experiments that manipulated the presence and predictability of non-communicative eye gaze behaviour in the lead-up to a joint attention episode. In both experiments, participants played a co-operative game where they needed to catch a burglar using nothing but their gaze (adapted from *Caruana, Brock & Woolgar, 2015*; also see *Caruana et al., 2017b* for an in-depth discussion and review of this methodological approach). The game was played with an avatar controlled by a gaze-contingent algorithm, and participants were asked to collaborate with their partner by initiating and responding to joint attention bids (see Fig. 1 for stimuli presentation). To do this, they both needed to search through their allocated row of houses to find the burglar. We refer to this portion of the trial as the 'search-phase'. Critically the task was programmed such that the avatar always completed their search for the burglar last. As such, during the search phase participants observed their partner make a series of non-communicative eye movements as they completed their search. At the end of the search phase, participants were required to initiate joint attention if they found the burglar, or respond to their partner's joint attention bid if they did not. Participants were not provided with any explicit instructions as to how they should perform the task, but for them to do it correctly they needed to establish eye contact before guiding or being guided by the avatar. We manipulated the presence and predictability of the avatar's non-communicative searching eye movements across two experiments and measured the time it took participants to respond to a guiding cue by making an accurate saccade to the correct house.

In the first experiment, the presence of irrelevant non-communicative gaze cues was manipulated. Here, we compared joint attention responsivity when the avatar displayed an irrelevant non-communicative (*i.e.,* Random Search) sequence of eye movements before establishing eye contact with a simpler version of the task where the search phase was removed (*i.e.,* NoSearch). In the second experiment, the informative nature of the non-communicative searching gaze pattern was manipulated to investigate its effect on joint attention responsivity. Here, we compared the same Random Search condition (described above) with a more closely-matched 'Predictive' search condition, in which the avatar's non-communicative search gaze was predictive of the target location (*i.e.,* Predictive Search). Specifically, the avatar was programmed to look at the burglar's location last before establishing eye contact and guiding the participant to that same location. This differs to the Random Search condition where the final gaze shift during the avatar's search was randomly determined. Participants also completed non-social versions of these tasks in which gaze cues were replaced with arrow cues. Even though previous studies indicate no reliable differences in response times in cueing studies comparing eye gaze and arrow cue conditions, it is important to note that these studies used non-interactive gaze-cueing paradigms (see meta-analysis for a review, *Chacón-Candia et al., 2022*). It was, hence, important to investigate if we find different effects in dynamic contexts that reflect the contexts in which we typically observe and respond to gaze information during dyadic, face-to-face interactions (*i.e.,* where the eye gaze of others is constantly changing in both informative and uninformative ways), and where the participants' goal is to intentionally

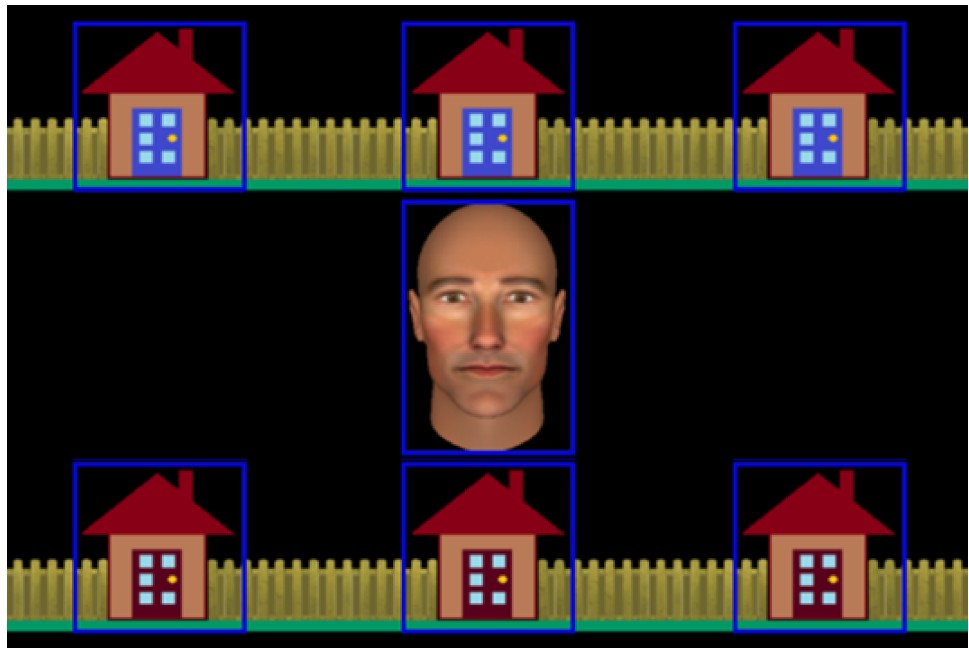

**Figure 1 Stimuli used in the interactive joint attention task.** The figure shows the central avatar and the six houses in which the burglar could be 'hiding'. Gaze-related areas of interest (AOIs), are represented by blue rectangles. These were not visible to participants.

respond to this spatial information. We found that people were faster to respond to joint attention bids when the non-communicative eye movements offered relevant (Predictive), rather than irrelevant (Random), spatial information about the locus of joint attention. We also found tentative evidence that humans may be able to discount irrelevant (Random) non-communicative gaze information more readily than they are for non-social spatial cues such as arrows in dynamic contexts (*Caruana et al., 2020*).

The examination of dynamic gaze contexts by *Caruana et al. (2020)* involved two distinct experiments with different participant samples. As such, from these data we are unable to directly compare performance between the NoSearch and Predictive Search contexts. This is critical in elucidating whether an informative non-communicative gaze context such as this leads to faster response times than no gaze context at all. Another limitation of our previous work was that the perceptual difference in the transition between the 'search' and 'response' phases of each trial were not completely matched across social and non-social conditions. Specifically, the green fixation point and green arrow stimulus in the non-social 'Arrow' condition did not match the difference in perceptual salience between averted and direct gaze stimuli in the social 'Eyes' condition. On Arrow trials, the green fixation point was followed by a green arrow that extended from the same green fixation point. This resulted in a relatively more subtle transition between 'searching' and 'guiding' arrow shifts since they were intervened by a fixation point of the same colour. This contrasts with the arguably more perceptually obvious transition between 'searching' and 'guiding' gaze shifts, which were intervened by direct gaze (*i.e.,* eye contact). As such, it is possible that the

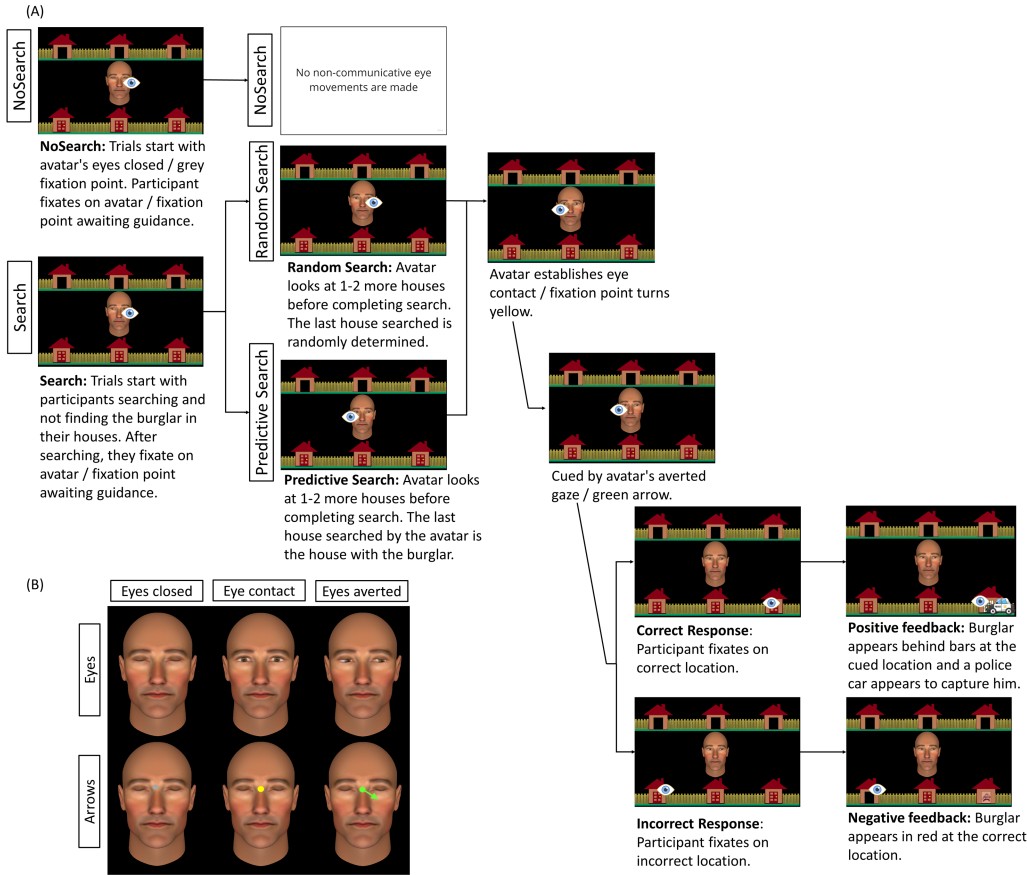

**Figure 2** **Schematic representation of trial sequence and central stimulus used by condition.** (A) Trial sequence for social Eyes stimuli for each of the three context conditions (NoSearch, Random Search, Predictive Search). The eye icon in the figure represents the fixation location made by the participants at each point in the trial and was not visible to them. Saccadic reaction time (SRT) was measured as the latency between cue presentation and the onset of responsive saccades. (B) An example of the central stimulus used in each condition, representing the analogous non-social stimulus used in the arrow condition for eyes closed, eye contact, and eyes averted. The background stimulus and the timing of stimulus presentation were identical across stimulus conditions (Eyes, Arrows).

previously-reported 'relative advantage' for eye gaze was not driven by a unique feature of eyes, but was rather driven by the more salient visual contrast between direct and averted gaze. To address this, we adapted the control stimulus in the current work, replacing the green fixation point stimulus with a yellow fixation point (see Fig. 2B).

## Current study

The current study endeavoured to rigorously examine how humans parse and are influenced by the non-communicative eye movements that precede eye contact and subsequent bids for joint attention. Building on recent work, we compared the three contexts (Predictive, No-Search, Random) using both stimulus sets (Eyes, Arrows) with a fully within-subjects design. Furthermore, to ensure the reported effects were indeed unique to the social interaction, we adapted the control stimulus originally implemented by *Caruana et al.*

*(2020)* to better match the perceptual changes observed during social trials. Based on our previous findings (*Caruana et al., 2020*), and consistent with the ostensive-inferential communication model (*Sperber & Wilson, 1986*; *Wilson & Sperber, 2002*), we hypothesised that even after implementing a more closely-matched control stimulus, ostensive signals (*i.e.,* eye contact) would result in a uniquely social advantage by assisting participants to selectively and rapidly respond to communicative gaze shifts. This would be characterised by participants showing slower responses on the Random Arrows condition but not the Random Eyes when compared to the NoSearch condition. Additionally, we expect participants to show faster responses on the Predictive Search condition when compared to NoSearch, with larger effects observed for Eyes than Arrows.

## MATERIALS & METHODS

### Ethics statement

All procedures implemented in this study were carried out in accordance with the protocol approved by the Macquarie University Human Research Ethics Committee (ID: 3775). Participants provided written, informed consent to take part in this study prior to participation.

### Participants

In line with previous studies using variations of the same task (*Caruana et al., 2020*; *Caruana et al., 2017a*), thirty-one adult participants were recruited from a pool of undergraduate psychology students at Macquarie University. The experiment lasted 1.5 h, with approximately 60 mins dedicated to completing the virtual interaction task. Participants were compensated with course credit for their time. All participants reported normal or corrected-to-normal vision (with clear contact lenses only), and reported no history of neurological injury or impairment. Two participants stated that they did not believe the deceptive cover story used in this study (see below) and were excluded from all analyses. The final sample included 29 participants (21 females: 8 males; $M_{age} = 19.76$ years; $SD = 4.4$).

### Stimulus and apparatus

Participants were seated at a table with a chin and forehead rest installed to stabilise head movements and standardise the screen viewing distance. Participants played a cooperative game with an on-screen avatar. The experimental stimuli were presented using Experiment Builder 1.10.165 on a 27-inch AOC monitor (display size: 59.8 cm × 33.6 cm; resolution: 1920 × 1080 pixels; refresh rate: 144 Hz) positioned 80 cm away from the participant. A remote desktop-mounted Eyelink 1000 (SR Research Ltd., Ontario, Canada) was used to record eye-movements from the right eye at a sampling rate of 500 Hz. Before starting each block, a nine-point eye-tracking calibration and validation was implemented.

During the task trials, an avatar, presented as an anthropomorphic animated face subtending 6.08° × 3.65° of visual angle, was displayed in the centre of the screen surrounded by six houses, each subtending 3.58° of visual angle (see Fig. 1). There were seven invisible gaze-related areas of interest (AOI) defined around the avatar's face
and each of the six houses that were used by our gaze-contingent algorithm during the experiment and for subsequent data processing (see *Caruana, Brock & Woolgar, 2015*, for detailed description of AOI definition).

## Design and procedure

Participants were told that they would be playing a collaborative game with two different members of the research team, named 'Alan' and 'Tony', who would be interacting with them from an adjacent laboratory. This cover story was important since previous studies using both more traditional gaze-cueing paradigms and variations of our own paradigm have shown differences in subjective experience (*Caruana, Spirou & Brock, 2017*), behavioural responses (*Caruana, Spirou & Brock, 2017*; *Morgan, Freeth & Smith, 2018*; *Teufel et al., 2010*; *Wiese, Wykowska & Müller, 2014*), and neurological responses (*Caruana & McArthur, 2019*; *Caruana, De Lissa & McArthur, 2017*) when participants believe their virtual partner is controlled by a human rather than a computer. Participants were told that their partner's eyes would be recorded using an eye-tracker and used to control the eye movements of the avatar they saw on their screen. Participants were also told that they would control an avatar presented on Alan/Tony's screen in the same way. In reality, the avatar's eye movements were controlled by a gaze-contingent algorithm that updated each time the participant moved their eyes. For example, every time the participant fixated a house to search for the burglar, the avatar stimulus was also updated to depict a shift in gaze (for a detailed description of the algorithm, see *Caruana, Brock & Woolgar, 2015*). This simulated the experience of a convincing and reciprocal social interaction.

Each trial began with the participant searching for the burglar in the houses with blue doors located at the top of the screen, while the avatar searched houses with red doors at the bottom of the screen. To search their allocated houses, participants had to look at a house before the door opened to either reveal the burglar or an empty house. Using the gaze-contingent algorithm, the avatar was designed to look at a different house each time the participant shifted their gaze between houses so that it appears to be searching for the burglar. To ensure that the avatar's search behaviour was realistically variable and that participants could not predict which or how many doors their partner would search before initiating joint attention, some trials started with one or two houses already open and empty. This helped introduce variability in the spatial sequence of participant's search behaviour, and in turn, justify the variability observed in the avatar's search behaviour. The sequence and number of open houses were systematically varied and counterbalanced across trials. This ensured that the avatar's search behaviour was realistically variable and that participants could not predict which or how many doors their partner would search before initiating joint attention.

On responding trials, participants would not find the burglar in any of their houses and had to wait for their partner to complete his search and guide them to the correct location. Once the participant completed their search and fixated on the avatar's face, it searched 1–2 more houses before establishing eye contact. This ensured that participants observed the avatar's non-communicative (*i.e.,* searching) and communicative (*i.e.,* guiding) eye gaze behaviours. This also allowed us to manipulate the presence and predictability of

non-communicative gaze cues displayed during this searching phase. On initiating trials, participants found the burglar in one of their houses and had to capture their partner's attention by establishing eye contact before guiding him to the correct location. Participants were told that for them to successfully catch the burglar they needed to both look at the target location. Participants were not given explicit instructions as to how they should guide or be guided by their partner. Once joint attention was achieved, the burglar appeared behind bars with a police car 'arriving' at the correct location to provide positive feedback.

In this study, we manipulated both the spatial context preceding the joint attention bid as well as the cue stimulus used to control for non-social attention effects, yielding six conditions: three *context* conditions (NoSearch, Random Search and Predictive Search) and two *stimulus* conditions (Eyes and Arrows). This resulted in each participant completing six condition blocks, with 30 responding trials and 30 initiating trials per block. Although the primary planned analyses for this study were for responding trials, it was essential to retain initiating trials in the task as they were critical for establishing the reciprocal joint attention task context.

The primary conditions were arranged into blocks, with block/condition order counterbalanced across participants. However, for each participant, the two stimulus conditions within each context condition were administered consecutively, in counterbalanced order, to minimise the switching between spatial cue contexts (*e.g.,* Eyes Random Search, Arrows Random Search; Eyes Predictive Search, Arrows Predictive Search; Eyes NoSearch, Arrows NoSearch). Within each condition block, trial order was randomised to ensure that the location of the burglar and the number of gaze shifts made by the avatar were not conflated with any potential order effects.

Participants could make four types of errors on each trial: (1) Search errors occurred when participants spent more than 3000 ms looking away from the avatar or their houses during the search-phase, which resulted in the text "Failed Search" appearing on the screen; (2) Timeout error occurred when participants took over 3000 ms to respond to the relevant gaze-cue; and (3) Location errors occurred when participants responded by looking at an incorrect location (see 'Results' for detailed breakdown of error data). For both Timeout and Location errors, the burglar appeared in red at the target location to provide negative feedback. Error trials were excluded from subsequent analyses of Saccadic Reaction Time (SRT) data.

## Context conditions

We manipulated the gaze information displayed during the non-communicative search-phase of two context conditions, (Random and Predictive) and a third 'NoSearch' baseline context condition that did not comprise any non-communicative information. See Fig. 2A for a summary of the trial sequence for each of these conditions.

### Random search condition

In this condition, when participants completed their search and were waiting for 'Alan' to establish eye contact and initiate joint attention, the final gaze shift made by Alan before establishing eye contact was randomly determined. This is consistent with the

original implementations of this task (*Caruana, Brock & Woolgar, 2015*; *Caruana et al., 2018*) which ensured participants could not predict where Alan was going to guide them based on the final target searched.

### Predictive Search condition

This condition was identical to the Random Search condition, with the exception that the final house searched by the avatar before establishing eye contact was always the target location, thus predicting the upcoming locus of joint attention. To prevent participants from detecting this systematic difference in avatar behavior across the Predictive and Random contexts, a cover story was implemented stating that participants would be playing the game with another member of the research team named 'Tony'. It is worth noting that participants were not explicitly informed of this manipulation. To verify whether participants detected any systematic difference between the Random and Predictive context conditions, we performed a post-experimental interview to collect descriptive information about their subjective experiences. Indeed, six people reported noticing the difference between Alan and Tony's search patterns. Their data was not excluded as our goal was to determine whether context effects manifested irrespective of whether participants consciously detected the variation in the search pattern or not. For a full summary of the subjective data, see documentation on our OSF Project page: https://osf.io/e7kg8/.

### NoSearch condition

In this condition, participants completed a simpler version of the task with Alan in which there was no need to search for the burglar on each trial. On initiating trials, participants could see only one blue door remaining closed, which they knew to conceal the burglar. On these trials, they simply waited for Alan to open his eyes to establish eye contact and initiated joint attention using a single gaze shift. Similarly on responding trials, Alan opened his eyes to establish eye contact and guided participants to the correct location using a single gaze shift.

## Stimulus conditions

A matched control condition was completed as separate blocks for each context condition. This was implemented to control for non-social task demands, such as attentional, oculomotor and inhibitory control. This resulted in two stimulus conditions (Eyes, Arrows) using the same gaze-contingent algorithm for presenting stimuli in both conditions. As such the spatial and temporal dynamics in these two conditions were identical. For the arrow condition, participants were informed that they were completing a computer-simulated version of the task in which a computer-controlled arrow stimulus was used to guide them to the correct location. The avatar's face remained on the screen with its eyes closed to best match the visual context across all conditions. At the beginning of each block, a grey fixation point subtending a visual angle of 0.29° was presented in between the avatar's eyes(analogous to the avatar closing its eyes on social gaze trials). The fixation point then turned yellow (analogous to the avatar displaying direct eye gaze). This was followed by a green arrow extending from a green central fixation point subtending a visual angle of 1.08°. Critically, this was analogous of the avatar averting its gaze. During the search phase

of the Random and Predictive Search conditions, the arrow stimulus was updated to point at different houses to match the avatar's searching behaviour within the social condition. Participants were informed that during the search phase on Arrow trials the arrow would randomly change direction. Once participants completed their search and looked back at the central area of interest (AOI), the arrow stimulus pointed at 1–2 more houses before being replaced by the yellow fixation point, analogous to the avatar making eye contact. This was then followed by a single green arrow pointing towards the target house which participants needed to follow to successfully 'catch the burglar'.

## Statistical analyses

Interest area and trial reports were exported using DataViewer software (SR Research Ltd., Ontario, Canada) to analyse the accuracy and Saccadic Reaction Time (SRT) data. For accuracy analyses, 'Calibration' and 'Search' errors were removed (128 trials total) before analysing the remaining trials for the proportion of correct trials. This was done because these errors occurred before the relevant gaze or arrow cue was presented and, hence, do not represent true joint attention errors. This resulted in analysing accuracy on 5,092 trials in total. For SRT analyses we excluded incorrect trials (491 trials total) and trials where participants responded faster than 150 ms (464 trials total), as these were likely to be anticipatory responses (*Carpenter, 1988*). No other outliers were identified. This resulted in the removal of 955 trials with 4,265 trials included in the SRT analyses. Raw eye-tracking data was screened and analysed in R using a custom script (https://osf.io/e7kg8/).

Consistent with our previous studies (*Caruana et al., 2020*; *Caruana et al., 2017a*) logistic and linear mixed random effects (LME) were used to analyse accuracy and SRT respectively. Specifically, we wanted to evaluate evidence for effects of Context, Stimulus and their interaction. The maximum likelihood estimation method was implemented in these analyses using the lme4 R package (*Bates & Sarkar, 2005*) and *p*-values were estimated using the lmerTest package (*Kuznetsova, Brockhoff & Christensen, 2015*).

We were interested in investigating the main effect of stimulus (Eyes, Arrows), context (Random, NoSearch, Predictive), and their interaction. To do this we used the 'successive differences contrast coding' method within the 'MASS' package in R (*Ripley et al., 2013*). This contrast method estimates effect parameters by sequentially comparing each level of context with the next level specified in the model. This method was used to provide parameter estimates for the overall effect of stimulus as well as the context effects between (1) NoSearch and Random Search, and (2) NoSearch and Predictive Search. Of critical interest in the current study, we were interested in testing the interaction of context and stimulus effects. As such, parameter estimates were also obtained for the context-by-stimulus interaction in (1) and (2) above.

We also tested the effect of context between the Predictive and Random Search conditions as well as their interaction with stimulus. However, this contrast could not be estimated using the predefined successive differences contrast coding method. This is because in this coding method the contrast coefficients are chosen so that the coded coefficients are the differences between the means of the second and first levels, the third and second levels, and so on. Therefore, we used the 'emmeans' package to manually define the missing

comparisons (*Lenth et al., 2019*). Finally, we ran post-hoc analyses for accuracy and SRT data to assist in interpreting significant stimulus-by-context interaction effects. An FDR correction was then applied to these post-hoc contrasts to confirm significance after correcting for multiple comparisons (*Benjamini & Hochberg, 1995*).

Accuracy and SRT models were defined with maximally-defined random-factor structures, including random intercepts for trial, block order and by-subject random slopes for the intercept and fixed effects (*Barr et al., 2013*). For SRT analyses, the residuals of the raw data violated the normality assumption and hence data were transformed using an inverse transformation. The normality assumption was confirmed after applying the transformation (details can be found in accompanying R code and output (https://osf.io/e7kg8/); see *Balota, Aschenbrenner & Yap, 2013*). All analyses had a significance criterion of $\alpha = 0.05$.

For estimating effect-size, the chi-squared goodness-of-fit tests were performed comparing a number of mixed random-effects models using Chi-square likelihood ratios to quantify the contribution of each fixed effect and interaction parameter to the model fit (*Johnston, Berry & Mielke Jr, 2006*). Unlike traditional measures of effect-size (*e.g.*, r2), this method provides an estimation of the variance explained by each fixed effect while accounting for variance independently explained by the specified random effects (see *Caruana et al., 2021*; *Caruana et al., 2019*). For full details pertaining to all of the analyses described above, refer to the accompanying R code (https://osf.io/e7kg8/).

## RESULTS

### Accuracy

Overall, only 9.30% of trials involved errors, with the majority being Location errors (5.29% of trials), followed by Search (2.45%) and Timeout errors (1.55%). Descriptive statistics of estimated fixed effect parameters are summarised in Table 1. Accuracy data for context and stimulus conditions are illustrated in Fig. 3.

Overall, participants made significantly more errors on the Random than NoSearch trials ($\beta = 1.90$, $z = 5.34$, $p < .001$). However, we found no evidence for an effect of stimulus ($\beta = -0.06$, $z = -0.19$, $p = 0.85$) or context between the Predictive and NoSearch contexts ($\beta = -0.66$, $z = -1.43$, $p = 0.15$). There was also no stimulus-by-context interactions between NoSearch and either the Random ($\beta = -0.83$, $z = -1.83$, $p = 0.07$) or Predictive ($\beta = -0.07$, $z = -0.13$, $p = 0.89$) conditions. We used emmeans to compare the accuracy outcomes in the Random and Predictive Search contexts (see Statistical Analysis above for justification). This revealed a significant context effect ($\beta = -2.40$, $z = -3.74$, $p < .001$) and stimulus-by-context interaction ($\beta = -0.89$, $z = -2.34$, $p = 0.02$). Post-hoc pairwise comparisons investigating this stimulus-by-context interaction between Random and Predictive contexts were then defined, correcting for multiple comparisons using a false discovery rate (FDR) correction. This revealed a significant effect of context between Random and Predictive with a larger effect for Arrows ($\beta = -1.65$, $z = -4.31$, $p < .001$) than Eyes ($\beta = -0.75$, $z = -2.06$, $p = 0.04$).

**Table 1  Estimated fixed effect parameters for Accuracy.**

| Fixed effect | β-coefficient | Standard Error (SE) | z-value | p-value |
|---|---|---|---|---|
| **Context** | | | | |
| NoSearch-Random | 1.857 | 0.348 | 5.335 | <.000*** |
| Predictive-NoSearch | −0.657 | 0.460 | −1.428 | 0.153 |
| Predictive-Random | −2.400 | 0.642 | −3.738 | <.000*** |
| **Stimulus** | | | | |
| Arrows-Eyes | −0.057 | 0.302 | −0.190 | 0.849 |
| **Stimulus*Context** | | | | |
| NoSearch-Random | −0.828 | 0.453 | −1.828 | 0.068 |
| Predictive-NoSearch | −0.066 | 0.499 | −0.133 | 0.894 |
| Random-Predictive | −0.894 | 0.383 | −2.338 | 0.019* |
| **Follow-up comparisons** | | | | |
| Random-Predictive (Arrows) | −1.647 | 0.382 | −4.311 | <.000***[a] |
| Random-Predictive (Eyes) | −0.753 | 0.365 | −2.062 | 0.039[a] |

Notes.
*$p < .05$
**$p < .01$
***$p < .001$
[a]Corrected $p$-values using a FDR correction for multiple comparisons.

**Table 2  Means (M) and standard deviations (SD) of SRT by Condition.**

| Condition | Random arrows | NoSearch arrows | Predictive arrows | Random eyes | NoSearch eyes | Predictive eyes |
|---|---|---|---|---|---|---|
| M (SD) | 470.90 ms (351.27) | 382.29 ms (227.21) | 338.65 ms (253.67) | 474.75 ms (344.56) | 445.04 ms (281.94) | 339.44 ms (196.38) |

Notes.
Means and standard deviations are provided in the format M(SD).

## Saccadic reaction times

Of critical importance in the current study, we investigated whether and how the presence of random and predictive spatial signals differentially influenced the speed with which participants were able to prepare saccadic responses to eye gaze and arrow cues. Mean SRT are summarised by condition in Table 2 and descriptive statistics of estimated fixed effect parameters are summarised in Table 3. All SRT data are summarised by context and stimulus conditions in Fig. 4.

Participants were significantly faster to respond during Predictive than NoSearch trials ($\beta = 0.64$, $t = 5.87$, $p < .001$). However, we found no evidence for a significant difference between the Random and NoSearch contexts ($\beta = 0.15$, $t = 1.98$, $p = 0.06$). We also found that, overall, participants were faster when responding to Arrows than Eyes, however this effect was marginally significant, since it was at our defined threshold for statistical significance ($\beta = -0.16$, t $= -2.02$, $p = .050$). Additionally, we found significant stimulus-by-context interactions when comparing each of the Random and Predictive contexts to the NoSearch context (Random: $\beta = -0.27$, t $= -4.08$, $p < .001$; Predictive: $\beta = 0.20$, $t = 2.91$, $p < 0.01$). The context effect and the stimulus-by-context interaction

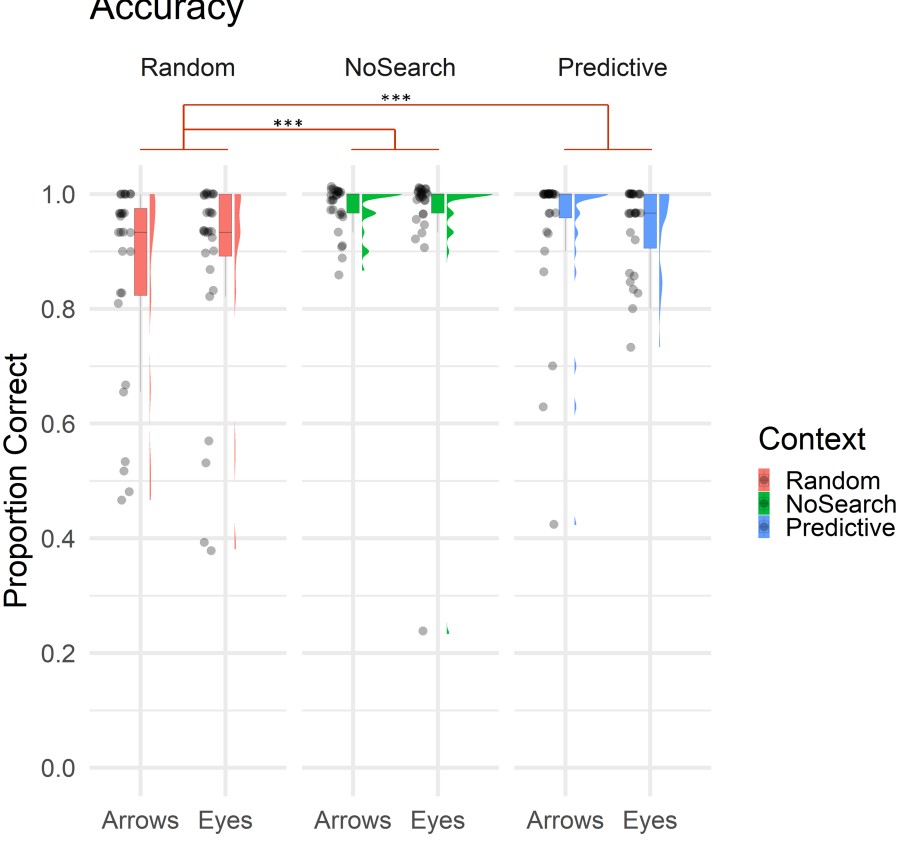

**Figure 3** Raincloud plot with individual data points illustrating the proportion of correct trials by context (Random Search, No Search, Predictive Search) and stimulus (Arrows, Eyes). In all boxplot figures, whiskers extend (as in a conventional Tukey's boxplot) 1.5 times the length of the box (*i.e.,* the interquartile range of the 1st and 3rd quartiles). Significant effects of context are illustrated in red (*** $p <$ .001).

between Random and Predictive contexts were also estimated using emmeans and revealed significantly slower responses on the Random context than the Predictive context ($\beta = -1.57$, t $= -8.78$, $p < .001$) but no significant interaction ($\beta = -0.07$, t $= -1.00$, $p = 0.32$). We used emmeans to conduct post-hoc pairwise comparisons of interest using custom contrasts, and correcting for multiple comparisons using a FDR correction, to help understand the significant interaction effects. This analysis revealed a significant effect of stimulus in the NoSearch context ($\beta = 0.31$, $t = 3.71$, $p = .001$) but not the Random ($\beta = 0.04$, $t = 0.46$, $p = 0.75$) or Predictive contexts ($\beta = 0.11$, $t = 1.27$, $p = 0.29$). We also found a significant effect of context between Random and NoSearch with Arrows ($\beta = -0.29$, t $= -3.45$, $p = .002$) but not Eyes ($\beta = -0.01$, t $= -0.16$, $p = 0.88$). Further, there was a significant effect of context between Predictive and NoSearch with a larger effect for Eyes ($\beta = -0.73$, t $= -6.48$, $p < .001$) than Arrows ($\beta = -0.53$, t $= -4.70$, $p < .001$).

**Table 3 Estimated fixed effect parameters for SRT.**

| Effect | $\beta$-coefficient | Standard Error (SE) | $t$-ratio | $p$-value |
|---|---|---|---|---|
| **Context** | | | | |
| NoSearch-Random | 0.149 | 0.076 | 1.978 | 0.058 |
| Predictive-NoSearch | 0.635 | 0.108 | 5.871 | <.000[***] |
| Predictive-Random | −1.568 | 0.179 | −8.783 | <.000[***] |
| **Stimulus** | | | | |
| Arrows-Eyes | −0.156 | 0.077 | −2.016 | 0.050[*] |
| **Stimulus*Context** | | | | |
| NoSearch-Random | −0.273 | 0.067 | −4.075 | <.000[***] |
| Predictive-NoSearch | 0.201 | 0.069 | 2.910 | 0.004[**] |
| Random-Predictive | −0.072 | 0.072 | −1.002 | 0.316 |
| **Follow-up comparisons** | | | | |
| NoSearch-Random (Arrows) | 0.286 | 0.083 | 3.446 | 0.002[**a] |
| NoSearch-Random (Eyes) | 0.013 | 0.083 | 0.158 | 0.875[a] |
| Predictive-NoSearch (Arrows) | 0.534 | 0.114 | 4.702 | <.000[***a] |
| Predictive-NoSearch (Eyes) | 0.735 | 0.113 | 6.477 | <.000[***a] |
| Arrows-Eyes (NoSearch) | 0.314 | 0.085 | 3.706 | .001[**a] |
| Arrows-Eyes (Random) | 0.041 | 0.088 | 0.464 | 0.752[a] |
| Arrows-Eyes (Predictive) | 0.113 | 0.089 | 1.265 | 0.294[a] |

Notes.
[*]$p < .05$
[**]$p < .01$
[***]$p < .001$
[a]Corrected $p$-values using a FDR correction for multiple comparisons.

### Model fit analyses

For quantifying the effects of stimulus and context, model-fit-improvement was compared as a function of each fixed effect parameter. Compared to the null model (*i.e.,* a model with no fixed-effect factors; AIC = 11647.49, BIC = 11730.14), adding the context factor significantly improved the model fit by 36.05 times (AIC = 11615.43, BIC = 11710.81, $X^2(1) = 36.05, p < 0.001$). Adding the stimulus factor to the context-only model improved the model fit another 5.50 times (AIC = 11611.94, BIC = 11713.67, $X^2(1) = 5.50, p = 0.019$). On the other hand, including the stimulus factor to the null model first enhanced the model's fit by only 5.71 times (AIC = 11643.78, BIC = 11732.79, $X^2(1) = 5.71, p = .017$), while adding the context effect to the stimulus-only model significantly improved the model fit 35.84 times (AIC = 11611.94, BIC = 11713.67, $X^2(1) = 35.84, p < .001$). Critically, compared to a model containing fixed-effect factors for both stimulus and context, adding the interaction parameter significantly improved the model fit by 20.09 times (AIC = 11595.85, BIC = 11710.30, $X^2(1) = 20.09, p < .001$). These analyses show a markedly larger effect of context than stimulus. However, it also suggests that both factors explain unique variance in the data and that the data are significantly better explained by a model that specifies a stimulus-by-context interaction.
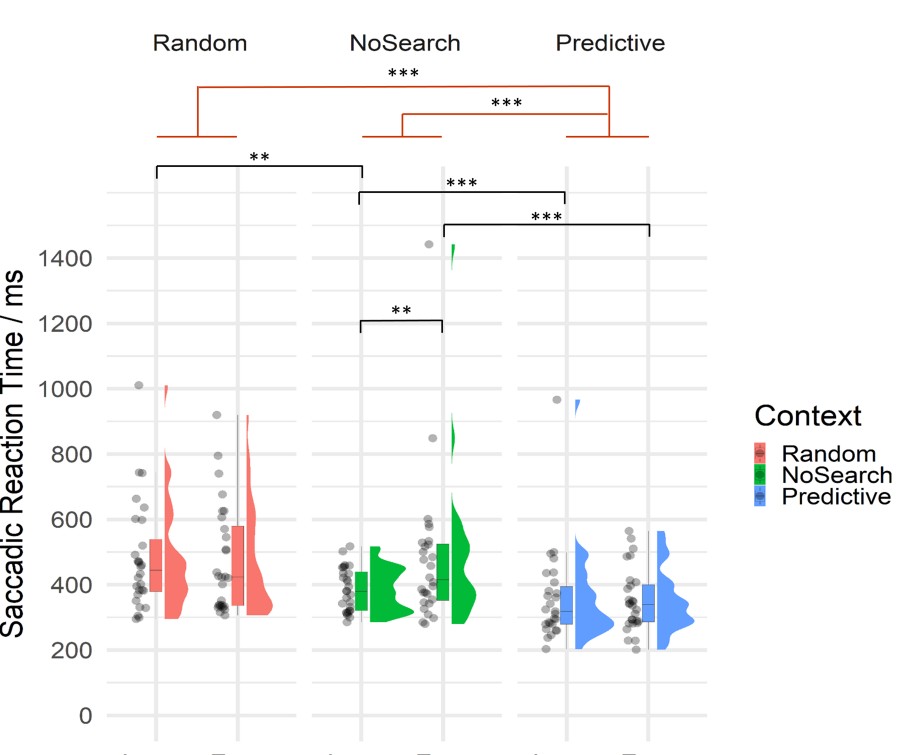

**Figure 4** **Raincloud plot with individual data points for saccadic reaction times on correct trials by context (Random Search, NoSearch, Predictive Search) and stimulus (Eyes, Arrows).** Significant effects of context are illustrated in red while post-hoc contrasts are shown in black (** $p < .01$, *** $p < .001$).

## DISCUSSION

Using a fully within-subjects design, we tested whether joint attention response times differ when following communicative joint attention bids that are preceded by: (1) **predictive** (*i.e.*, informative) non-communicative eye movements; (2) **random** (*i.e.*, uninformative) non-communicative eye movements; or (3) **no eye movements at all**. We further examined whether any observed differences are specific to the social domain by comparing performance in an analogous task in which eye movement cues were replaced with dynamic arrow stimuli. We found that participants did effectively track and use predictive information embedded in other's eye movements before a joint attention bid; characterized by faster joint attention responses following **predictive** rather than **no** non-communicative eye movement sequences. Whilst similar patterns were observed for **arrow** stimuli, we present evidence for a relative advantage for utilizing *predictive* information embedded in **eye** movements. This was further supported by the finding of a larger cost of observing *random* spatial information embedded in dynamic **arrow** sequences, but not for **eyes**. This demonstrates that participants were better able to disregard uninformative spatial information when it was conveyed as non-communicative **eyes** than **arrows**.

### Context effects (random *vs* predictive *vs* nosearch)

In this study, we found that participants were faster to respond within the Predictive context than both the NoSearch and Random contexts, irrespective of the task stimuli used (Eyes or Arrows). That is, we observed slower reaction times in the Random than Predictive condition, even though the contexts were matched for their complexity and dynamics. We also observed slower reaction times for the NoSearch than Predictive condition, even though the Predictive context was arguably more complex, with additional demands on attention. Together, these findings highlight our capacity to extract and use predictive social and non-social spatial cues to optimise attention allocation. This aligns with claims within the Relevance Theory, that cognition is geared towards allocating attention to information that is maximally relevant in a given context (*Sperber & Wilson, 1986*). The notion of maximal relevance refers to attending to and processing information that produces the greatest cognitive effect for the least effort when compared to any other inputs available at the time. In the context of our task, when relevant information about the location of the burglar was available, participants were sensitive to it and used it to enhance their responsivity, compared to contexts with only irrelevant information or no information at all.

Interestingly, reaction times in the Random context did not significantly differ from the NoSearch context. This contrasts with our earlier findings, where we reported slower responses on the Random context compared to the NoSearch context (*Caruana et al., 2020*). This non-replicated effect may be attributed to the implementation of a more closely-matched non-social stimuli, which in the current study made it easier to perceptually distinguish between irrelevant and relevant non-social arrow cues in the Random condition (discussed further below).

### *Context by stimulus interactions*

Although we found faster responses to Arrows than Eyes when no non-communicative eye movements were present (*i.e.,* NoSearch condition, discussed further below), we critically observed that a relative gaze advantage emerged when participants were required to respond to Eyes in dynamic contexts (*i.e.,* Random and Predictive contexts) relative to the NoSearch baseline condition. Specifically, when comparing Random and NoSearch conditions, we observed a specific disadvantage for Random trials when responding to Arrows but not Eyes. Further, when comparing Predictive and NoSearch trials, we observed a relatively larger advantage for Eyes than Arrows. Together, these findings suggest that people are better positioned to *adaptively use* non-communicative gaze information when it offers opportunities to facilitate joint attention, and *disregard* this information when it does not, than they are for non-social spatial cues in analogous task contexts.

Existing evidence from studies using traditional gaze-cueing paradigms that investigate more reflexive gaze orientation have failed to identify reliable differences in responses between gaze and arrows cue conditions (see meta-analysis for a review, *Chacón-Candia et al., 2022*). Our study, however, presents evidence for a specific advantage for gaze when joint attention bids were embedded in a more naturalistic and dynamic stream of information. There are a few possible explanations for this gaze advantage. First, it is

possible that our rich experiences in gaze-based communication and social interaction have led to the development of expertise in evaluating the relevance of eye movements and rapidly discerning whether they are communicative, informative or irrelevant. As suggested by *Caruana et al. (2020)*, this expertise in evaluating the relevance of non-communicative social behaviour from gaze might not readily generalise to non-social stimuli. Indeed, in the current study, participants may have exhibited a relative disadvantage for responding to arrows in the Random context because they have less experience disregarding irrelevant arrow cues. They may have also exhibited a smaller relative advantage for responding to Arrows in the Predictive context than Eyes due to the lack of experience in parsing dynamic arrow sequences. This is unsurprising since arrows are rarely, if ever, irrelevant or presented in dynamic contexts. By contrast, humans have continuous exposure to social interactions involving a steady stream of mostly non-communicative eye movements displayed by interlocutors. A second explanation for this gaze advantage might be that humans have an innate endowment to evaluate and associate ostension to non-verbal social gestures such as eye contact (*Sperber & Wilson, 1986*; *Wilson & Sperber, 2002*). Social interaction requires the ability to distinguish between communicative and non-communicative gestures, which involves identifying communicative intent. This was afforded in the social condition given that non-communicative and communicative eye movements were intervened by eye-contact; an ostensive signal for communicative intent. By contrast, the arrows presented in the non-social control task required participants to differentiate between relevant and irrelevant cues, but did not include the identification of communicative intent. The presence of a relative social advantage for Eyes over Arrows suggests that identifying communicative intent from eye-contact facilitated more efficient joint attention responsivity. This might be, in part, because eye contact is an inherently potent social signal that rapidly activates cortical mechanisms involved in making inferences about others' intentions (*Burra, Mares & Senju, 2019*; *Conty et al., 2007*; *Kampe, Frith & Frith, 2003*; *Mares et al., 2016*; *Schilbach et al., 2006*; *Senju & Johnson, 2009*). Indeed, eye contact, especially during live interactions, induces psychophysiological arousal and increased alertness and attention (*Gale et al., 1975*; *Helminen, Kaasinen & Hietanen, 2011*; *Hietanen, Peltola & Hietanen, 2020*; *Kleinke & Pohlen, 1971*; *Nichols & Champness, 1971*). This explanation also aligns with past work using gaze cueing paradigms which have shown faster oculomotor response after the observation of eye contact (*Dalmaso et al., 2020*; *Xu, Zhang & Geng, 2018*; *Kompatsiari et al., 2018*). More research is needed to further probe the features of eye contact that can further facilitate the perception of communicative intent during dynamic interactions, by manipulating variables such as the presence, frequency and duration of eye contact in the lead up to joint attention opportunities. The knowledge gained from such studies will inform how we can engineer optimally communicative signals in the design of artificial agents for human social interaction (*e.g.*, robots and virtual characters).

## Stimulus effect (eyes *vs* arrows)

When examining the overall effects of stimulus, we did not find a significant difference in reaction times between Eyes and Arrows within the Random or Predictive contexts. This differs from our earlier findings in which we observed faster responses to Arrows compared

to Eyes within the Predictive context (*Caruana et al., 2020*). As previously mentioned, one key difference between the current study and *Caruana et al. (2020)* is the modification of our non-social stimuli to better match the perceptual change in stimuli that occurs when the avatar established eye contact (*i.e.,* averted gaze [search] → **eye contact** → averted gaze [cue]). The non-social arrow and fixation point stimulus used in our previous studies presented an arguably less obvious transition between the search and response phase (*i.e.,* green fixation point + green arrow [search] → **green fixation point** → green fixation point + green arrow [cue]) compared to the ostensive eye contact stimulus in the social condition. As such, it is possible that we previously saw slower reaction times to eye gaze in the Predictive context because the eye contact stimulus was more visually salient than the fixation point. Thus, participants may have taken longer to disengage and respond to the subsequent joint attention bid. The current study, however, modified the non-social stimuli by changing the colour of the fixation point that was analogous to eye contact, to provide a better visual match across the Eye and Arrow conditions. Our current results indicate that this did indeed provide a better match for the perceptual salience of stimuli, given that we observed more commensurate response time for both Predictive Eyes and Arrows.

It is noteworthy, however, that despite implementing more perceptually matched stimuli in the current study, we still find evidence for significantly faster reaction times to Arrows than Eyes in the NoSearch task, consistent with our previous findings. Previously, we discussed the possible role of experience in parsing and responding to Eye and Arrow spatial cues in dynamic contexts (*i.e.,* Random and Predictive contexts). In the same way that we may have more experience responding to relevant eye cues than arrows in *dynamic* contexts, it is also likely the case that most people have more experience observing and responding to *static* arrows than eyes in real-world contexts. This might explain why we see faster responses to Arrows than Eyes in the NoSearch task where non-dynamic spatial cues were presented to participants. Alternatively, it is also possible that slower responses to eyes in the NoSearch condition are due to eye contact's cascading effect on downstream attention and social cognition processing (*Dalmaso, Castelli & Galfano, 2017*; *Gale et al., 1975*; *Helminen, Kaasinen & Hietanen, 2011*; *Hietanen, Peltola & Hietanen, 2020*; *Kleinke & Pohlen, 1971*; *Nichols & Champness, 1971*). This may increase cognitive processing load by automatically engaging higher-order social-cognitive processing (*Burra, Mares & Senju, 2019*; *Conty et al., 2007*; *Kampe, Frith & Frith, 2003*; *Mares et al., 2016*; *Schilbach et al., 2006*; *Senju & Johnson, 2009*). As such, the presence of direct gaze during non-dynamic tasks may contribute to slower responses to Eyes than Arrows due to slower disengagement from direct gaze stimuli. This is uniquely seen in the NoSearch context where the trial begins with eye contact—adding to its salience—and where eye contact does not serve to differentiate communicative from non-communicative gaze. As demonstrated in the current study, characterizing this fundamental difference in responding to stand-alone gaze and arrow stimuli in non-dynamic contexts (*i.e.,* the NoSearch condition) offers a useful 'baseline' that can help interpret differences in stimuli, or the lack of, when interpreting additional differences that may emerge in more dynamic contexts (*e.g.,* Random and Predictive conditions). The current findings also demonstrate the potential inadequacies

of previous gaze following paradigms of gaze processing and joint attention that have presented participants with a single gaze cue on each trial (see *Caruana et al., 2017b* for review), given that the NoSearch context—which most closely resembles this common approach—produced very different stimulus effects to those observed in the more dynamic, and ecologically-valid contexts (*e.g.*, Random, Predictive).

### Implications and future application

The current findings, and our updated paradigm, offers a new tool for carefully examining social information processing in dynamic contexts. Our findings provide a valuable starting point for constructing a relevance model of gaze-based joint attention that critically considers the dynamic contextual properties of gaze that shape social perception and action. This is of key theoretical importance given that our understanding of gaze perception to date has been built on findings from experimental work using non-interactive paradigms, in which gaze stimuli are presented in non-dynamic contexts. A new relevance model of gaze-based joint attention that accounts for the dynamic properties of gaze—and paradigms that can test these properties systematically—are critical for advancing our understanding of gaze use during realistic social interactions with artificial agents and between neurodiverse humans.

For instance, this model can be instrumental in guiding research examining the factors that influence how humans perceive and respond to gaze information during interactions with artificial agents (*e.g.*, virtual characters and social robots). Such research is critical for taking a human-centered approach to the design of artificial agents that can achieve intuitive interactions with humans. Critical to this future empirical endeavor is understanding the extent to which social cognitive processes (*e.g.*, mental state attribution) have a top-down influence on the processing of communicative gaze signals during dynamic interactions. Future studies could begin to explore this using our paradigm by manipulating whether participants believe the avatar or arrow stimuli are being controlled by a human partner or computer. This would help elucidate, for instance, the extent to which mental state attribution underpins the relative advantage for eye gaze observed in dynamic contexts.

In moving towards more ecologically-valid conceptualizations of joint attention, our approach also offers opportunities for studying joint attention difficulties experienced by autistic people (*e.g.*, *Charman, 2003*; *Hobson & Hobson, 2007*; *Mundy, Sigman & Kasari, 1994*), who often report marked subjective difficulty in establishing eye contact with others (*Trevisan, Hoskyn & Birmingham, 2018*; also *Adrien et al., 1993*; *Kanner, 1943*; *Mirenda, Donnellan & Yoder, 1983*; *Zwaigenbaum et al., 2005*). Specifically, our paradigm can be used to interrogate why—and under what conditions—autistic individuals may experience difficulties responding to gaze-cued joint attention bids. Early studies have suggested this may be due to a difficulty in differentiating communicative and non-communicative gaze information (*Caruana et al., 2018*). The current paradigm would enable this to be directly interrogated to better understand the unique challenges that some autistic individuals face in processing social cues.

## CONCLUSIONS

This study provides evidence that people adaptively *use* and *dismiss* non-communicative gaze information depending on whether it informs the locus of an upcoming joint attention bid. This adaptive ability is not observed to the same extent when people process dynamic arrow stimuli in an analogous task which carefully controls for attention and perceptual task demands. Our findings have important implications for informing theoretical models of gaze processing in dynamic interactive contexts; highlighting the relative advantage people have for extracting others' relevant and predictive non-communicative gaze information. Our experimental approach using carefully controlled yet interactive paradigms also offers a useful tool for advancing the study of social information processing across neurodiverse people and with artificial agents.

## ACKNOWLEDGEMENTS

We would like to take this opportunity to thank Dr. Peter Humburg for his statistical support. We also thank the Technical Research and Innovation Support team, especially Marcus Ockenden and Craig Richardson, for their continued technical support.

### Funding

This research was supported by an Australian Government Research Training Program (RTP) Scholarship. Dr Nathan Caruana was supported by a Macquarie University Research Fellowship (MQRF). The funders had no role in study design, data collection and analysis, decision to publish, or preparation of the manuscript.

### Grant Disclosures

The following grant information was disclosed by the authors:
Australian Government Research Training Program (RTP) Scholarship.
Macquarie University Research Fellowship.

### Competing Interests

Nathan Caruana is an Academic Editor for PeerJ

### Author Contributions

- Ayeh Alhasan conceived and designed the experiments, performed the experiments, analyzed the data, prepared figures and/or tables, authored or reviewed drafts of the article, and approved the final draft.
- Nathan Caruana conceived and designed the experiments, analyzed the data, authored or reviewed drafts of the article, and approved the final draft.

## Human Ethics

The following information was supplied relating to ethical approvals (*i.e.*, approving body and any reference numbers):

The Macquarie University Human Research Ethics Committee granted ethical approval to carry out the study within its facilities (Ethical Application Ref ID: 3775)

## Data Availability

The full data set and R code with the analysis outputs and annotated code descriptions are available on the Open Science Framework: Alhasan, Ayeh, and Nathan Caruana. 2023. "Evidence for the Adaptive Parsing of Non-Communicative Eye Movements during Joint Attention Interactions." OSF. October 4. doi: 10.17605/OSF.IO/E7KG8.

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
