# Peer review of "Evidence for the adaptive parsing of non-communicative eye movements during joint attention interactions"

_PeerJ, doi:10.7717/peerj.16363_

## Round 0.1 · original submission · Minor Revisions

Looking over the reviewer comments, I note that all comments are quite minor. Most are simply requesting some further little clarification across different places in the paper. I encourage the authors to go through the comments and provide the clarifications. I can see how adding these clarifications will improve the paper. Any reviewer comments that the authors do not agree with, please provide a rebuttal as to why edits were not made. From what I can see there doesn't appear to be anything I expect would be of any issue.

Reviewer 1 ·

Basic reporting

The work is written well, and the introduction is comprehensive and informative. The study itself is ambitious and well executed. I am familiar with the previous work and this is an interesting and appropriate follow up. The research question is clear, relevant and timely. The study is well reported and the materials and data are open which is good.

Experimental design

The work is original and the research question is well defined and clear. The analysis seems comprehensive and appropriate. Methods section is clear and comprehensive.

I did have a couple of issues with regards to reporting here:
1. There is no clear sample size justification.
2. Line 206, page 10 – what is meant by responded appropriately – please provide detail.
3. related to 2, more needs to be explained about the algorithm used to create realistic gaze.
4. How long did participants spend in the study?

Validity of the findings

The conclusions are appropriate and measured. data is provided and stats seem valid.

I do have one issue with the conclusions in terms of interpretation of the arrow cue. I think there is a further possible way to interpret the arrow cue with respect to the instructions/ framing the participant is given around the arrow cue. Participants are told that the ‘computer-controlled arrow stimulus was used to guide them to the correct location’. This gives the arrow a much stronger meaning than the gaze which is supposed to just be another person looking at the stimuli. It therefore makes it possible that the participants saw the arrow as more meaningful than the gaze, leading to the reaction times differences and the seemingly unadaptive use of the arrow. This makes me wonder what would the participants have done if they believed the eyegaze was the computer (or another person) explicitly guiding them, or if they were told the arrow was random? This issue needs to be discussed, and should be considered for further study.

·

Basic reporting

Overall, this manuscript clearly provides enough details of the study. Figures are helpful in understanding the task. All raw data is supplied online including R code.

Experimental design

This study aimed to examine how the informativeness of another’s non-communicative eye movements influences joint attention using a gaze-contingent paradigm. There were two independent factors, the factor of context (Predictive, No-Search, Random) and stimulus (Eyes, Arrows), with a within-subjects design. The authors adapted the stimuli and task originally implemented by Caruana et al. This gaze-contingent paradigm of implicit joint attention would be useful to investigate spontaneous joint attention.

Validity of the findings

Line 173~ Participants
・Have you conducted sample size estimation?
・What was the gender ratio of the participants? It should be clearly reported since some studies have reported gender differences in gaze sensitivity.

Line 277~ 
・The authors mentioned that 6 people reported noticing the difference between Alan’s and Tony’s search patterns. Have you conducted analyses excluding these 6 participants?

Line 310~ Statistical analyses
・How many trials were excluded from the analyses?

Line 414~ Model fit analyses
・Can you present data on model fitting such as the AIC and BIC? To compare the model fitting levels, each model's index of fitting should be reported.

Lines 485-489
・The authors mentioned that experience may modulate responses to gaze cues. However, participants were faster to respond to Arrows than Eyes in general (Lines 396-397). It may be hypothesised that arrow cues cannot induce rapid responses if the amount of the experiences influences saccades. The authors should mention the different mechanisms of attention orientation between arrow- and gaze-cueing situations rather than discussing how experience modulates attentional responses.

Lines 528-530
Although the authors mentioned the limitation that the eye contact stimulus was more visually salient than the fixation point, in that case, other results of the difference between Arrow and Eyes also have to be more moderate. Again, I recommend discussing the difference in attention orientation mechanisms between Arrow and Gaze cues.

Additional comments

Thank you for providing me the opportunity to review this interesting study. Especially, the gaze-contingent paradigm used in this study has potential interest for researchers in social cognition. Since this task can measure spontaneous joint attention, it would be interesting to use the same task in people with ASD.
As mentioned above, I just have some concerns about the statistics and discussion. It would be challenging to integrate the current results, however, I would like to see the authors’ original theoretical contributions. The current discussion seems to explain each result separately rather than emphasising the theoretical contributions of this study.

Reviewer 3 ·

Basic reporting

This paper illustrates an eye-tracking study in which participants were engaged in a gaze-contingency task with an avatar (social condition) or a dynamic arrow (non-social condition). The main results show that the participants were influenced by the gaze of the avatar when looking for a target, and this was particularly evident when the gaze of the avatar was informative about the location of the target. In addition, these results were much more evident in the social than in the non-social condition.

I have appreciated this paper, which is well written and organized. I only proposed some comments aimed at increasing some theoretical motivations underlying this work.

1) Introduction

- Line 50-51: there are a couple of more recent and relevant reviews on this topic that I’m missing (Dalmaso, Castelli, et al., 2020; McKay et al., 2021)
- I have appreciated the fact that you also introduced the control non-social conditions with arrows, which is important to establish whether a given phenomenon is truly social or not; however, I feel that the theoretical motivation for this control condition should be better motivated; for instance, there is a recent meta-analyses showing that gaze and arrow seems to lead to vert similar results in social attention tasks, at least at the behavioral level (Chacón-Candia et al., 2022)
- Related to my previous comment, I’m wondering why in the control ‘non-social’ condition you decided to maintain the picture of the face with closed eyes; I can understand the advantage of presenting participants with a similar perceptual stimulus, but at the same time I’m wondering if this condition can be actually labelled as truly ‘non-social’

Methods
- Line 201-202: you primed participants with a cover story saying that it was an interactive game with another person placed in another laboratory; I’m wondering if this was necessary to observe the results you reported here; do you think that a different pattern could emerge if no cover story was provided? This manipulation reminds me of some work showing that ‘mental state’ attribution can shape responses to social attention responses (e.g. Wiese et al., 2014)
- Please clarify how the sample size was established (e.g., power analysis?); I am wondering if N = 31 (which seems relatively small) is enough to detect the effects that you wanted to examine.
- RTs analyses: you discarded trials with an RT < 150 ms (i.e., anticipations); what about trials with an excessive latency (i.e., outliers? If any). Please clarify.

Results
- As 6 participants reported differences between ‘Alan’ and ‘Tony’, I am wondering if these people have been included in the final analyses.
- I think it would be better to report p < .001 instead of p <.000
- Line 397: technically speaking, p = .05 is a non-significant result; please verify.

Discussion
- Line 508-509: I am missing another relevant work showing that eye contact can also increase the oculomotor response in a gaze cueing task (Dalmaso, Alessi et al., 2020)
- Lines 542-546: Please note that there is indeed evidence for the notion that direct-gaze stimuli can hold attention as reflected in oculomotor indexes; please see (Dalmaso et al., 2017) which could be briefly mentioned.


References
Chacón-Candia, J. A., Román-Caballero, R., Aranda-Martín, B., Casagrande, M., Lupiáñez, J., & Marotta, A. (2022). Are there quantitative differences between eye-gaze and arrow cues? A meta-analytic answer to the debate and a call for qualitative differences. Neuroscience & Biobehavioral Reviews, 104993. https://doi.org/10.1016/j.neubiorev.2022.104993

Dalmaso, M., Alessi, G., Castelli, L., & Galfano, G. (2020). Eye contact boosts the reflexive component of overt gaze following. Scientific Reports, 10(1), 4777. https://doi.org/10.1038/s41598-020-61619-6

Dalmaso, M., Castelli, L., & Galfano, G. (2017). Attention holding elicited by direct-gaze faces is reflected in saccadic peak velocity. Experimental Brain Research, 235(11), 3319–3332. https://doi.org/10.1007/s00221-017-5059-4

Dalmaso, M., Castelli, L., & Galfano, G. (2020). Social modulators of gaze-mediated orienting of attention: A review. Psychonomic Bulletin & Review, 27(5), 833–855. https://doi.org/10.3758/s13423-020-01730-x

McKay, K. T., Grainger, S. A., Coundouris, S. P., Skorich, D. P., Phillips, L. H., & Henry, J. D. (2021). Visual attentional orienting by eye gaze: A meta-analytic review of the gaze-cueing effect. Psychological Bulletin, 147(12), 1269–1289. https://doi.org/10.1037/bul0000353

Wiese, E., Wykowska, A., & Müller, H. J. (2014). What we observe is biased by what other people tell us: beliefs about the reliability of gaze behavior modulate attentional orienting to gaze cues. PloS One, 9(4), e94529. https://doi.org/10.1371/journal.pone.0094529

Experimental design

Please see my previous comments

Validity of the findings

Please see my previous comments

---

## Round 0.2 · accepted · Accept

Thank you for making changes, I believe the manuscript has been improved.

---

## Author Rebuttal · Round 0.2

**Response to Reviews:** Manuscript Ref# 2023:06:86928:0:1:REVIEW

Re: Manuscript **Ref# 2023:06:86928:0:1:REVIEW** *"Evidence for the adaptive parsing of non-communicative eye movements during joint attention interactions".*

Dear Dr Shane Rogers,

We appreciate the constructive feedback provided by the reviewers on our manuscript. We are pleased to acknowledge that all the comments raised by the reviewers have been carefully considered and addressed in the revised manuscript. We believe that the revisions have improved the quality of this manuscript. In this response letter, we outline the changes made in response to each reviewer's comments.

For your convenience, we have included excerpts of the requested revisions in this document in **red text**. Please note that the reviewers' original comments are in **blue text**, and our responses are in **black text**.

We have uploaded the revised manuscript, with tracked changes. We have also uploaded a .pdf copy of this response letter so that the colour scheme described above can be seen.

Warm Regards,

Ayeh Alhasan on behalf of all authors

**Response to Reviewer's Comments and Recommendations:**

**Reviewer 1 (Anonymous)**

**Basic reporting**

The work is written well, and the introduction is comprehensive and informative. The study itself is ambitious and well executed. I am familiar with the previous work and this is an interesting and appropriate follow up. The research question is clear, relevant and timely. The study is well reported and the materials and data are open which is good.

We would like to thank you for your positive feedback and thoughtful evaluation of our manuscript.

**Experimental design**

The work is original and the research question is well defined and clear. The analysis seems comprehensive and appropriate. Methods section is clear and comprehensive.

I did have a couple of issues with regards to reporting here:
1. There is no clear sample size justification.

We based our sample size on multiple previous studies using similar paradigms across samples of 25-30 participants. In these studies, we reported large effect sizes of context (β=2.017, β=0.890). It is worth noting, however, that these previous studies used a between-subjects design whereas our current study employed a more powerful, fully within-subjects design. As such, the target sample size of 30, which is at the higher end of our previous studies' sample sizes, was expected to provide a conservative estimate given that the current study employed a more powerful within-subjects design.

We have now added the following justification to lines (182-183):

> *"In line with previous studies using variations of the same task (Caruana et al., 2020; Caruana et al., 2017a), thirty-one adult participants were recruited from a pool of undergraduate psychology students at Macquarie University."*

2. Line 206, page 10 – what is meant by responded appropriately – please provide detail.

We have now clarified this statement on lines (223-225)

> *"In reality, the avatar's eye movements were controlled by a gaze-contingent algorithm that updated each time the participant moved their eyes. For example, every time the participant fixated a house to search for the burglar, the avatar stimulus was also updated to depict a shift in gaze (for a detailed description of the algorithm, see Caruana et al., 2015)."*

3. related to 2, more needs to be explained about the algorithm used to create realistic gaze.

We have made the following changes to clarify to lines (231 – 238):

> *"Each trial began with the participant searching for the burglar in the houses with blue doors located at the top of the screen, while the avatar searched houses with red doors at the bottom of the screen. To search their allocated houses, participants had to look at a house before the door opened to either reveal the burglar or an empty house. Using the gaze-contingent algorithm, the avatar was designed to look at a different house each time the participant shifted their gaze between houses so that it appears to be searching for the burglar. To ensure that the avatar's search behaviour was realistically variable and that participants could not predict which or how many doors their partner would search before initiating joint attention, some trials started with one or two houses already open and empty. This helped introduce variability in the spatial sequence of participant's search behaviour, and in turn, justify the variability observed in the avatar's search behaviour. The sequence and number of open houses were systematically varied and counterbalanced across trials."*

We have also directly referred readers to earlier work where the algorithm, and its development is described in extensive detail, as outlined in response to 2.

We have provided extra details as follows to line (184-185):

> *"The experiment lasted 1.5 hours, with approximately 60 mins dedicated to completing the virtual interaction task. Participants were compensated with course credit for their time."*

**Validity of the findings**

The conclusions are appropriate and measured. data is provided and stats seem valid.

I do have one issue with the conclusions in terms of interpretation of the arrow cue. I think there is a further possible way to interpret the arrow cue with respect to the instructions/ framing the participant is given around the arrow cue. Participants are told that the 'computer-controlled arrow stimulus was used to guide them to the correct location'. This gives the arrow a much stronger meaning than the gaze which is supposed to just be another person looking at the stimuli. It therefore makes it possible that the participants saw the arrow as more meaningful than the gaze, leading to the reaction times differences and the seemingly unadaptive use of the arrow. This makes me wonder what would the participants have done if they believed the eye gaze was the computer (or another person) explicitly guiding them, or if they were told the arrow was random? This issue needs to be discussed, and should be considered for further study

Thank you for this interesting comment. First, we would like to clarify that participants were told that the arrow would update randomly during the search phase. This was to mitigate any perceptions of the arrow reflecting some form of 'AI system'. This is a critical detail that we have now clarified in the manuscript under the **Stimulus Conditions** section lines (330-331):

> *"During the search phase of the Random and Predictive Search conditions, the arrow stimulus was updated to point at different houses to match the avatar's searching behaviour within the social condition.* *Participants were informed that during the search phase on Arrow trials the arrow would randomly change direction.* *Once participants completed their search and looked back at the central area of interest (AOI), the arrow stimulus pointed at 1-2 more houses before being replaced by the yellow fixation point, analogous to the avatar making eye contact. This was then followed by a single green arrow pointing towards the target house which participants needed to follow to successfully 'catch the burglar'."*

Second, we are very interested in your thoughts about how behaviour in this task context might play out if participants approached both stimulus sets under the same intentional stance or belief about the stimulus' agency. This is certainly something we have examined in our previous work using this paradigm. Earlier work has shown that intentional stance beliefs do influence subjective experience (Caruana, Spirou, et al., 2017), behavioural responses (Caruana, Spirou, et al., 2017; Morgan et al., 2018; Teufel et al., 2010; Wiese et al., 2014), and neurophysiological responses (Caruana & McArthur, 2019; Caruana, de Lissa, 2017) during this and similar gaze-related tasks. However, a past

study we conducted using this joint attention task found no differences in response times between people who believed the avatar was controlled by a human or computer. Nevertheless, we think there would be value in examining this further in future studies to see if this holds true when testing the specific context effects probed in this study, given that our previous work only examined this belief manipulation when completing the Random context of this task. In particular, we think this may help determine the extent to which mental-state attribution drives the relative gaze advantage observed in the dynamic contexts probed in this study. We have now added this future direction to our **Discussion** section lines (620-626).

> *"Critical to this future empirical endeavor is understanding the extent to which social cognitive processes (e.g., mental state attribution) have a top-down influence on the processing of communicative gaze signals during dynamic interactions. Future studies could begin to explore this using our paradigm by manipulating whether participants believe the avatar or arrow stimuli are being controlled by a human partner or computer. This would help elucidate, for instance, the extent to which mental state attribution underpins the relative advantage for eye gaze observed in dynamic contexts."*

**Reviewer 2 (Mitsu Ishikawa)**

**Basic reporting**

Overall, this manuscript clearly provides enough details of the study. Figures are helpful in understanding the task. All raw data is supplied online including R code.

Thank you for your positive feedback.

**Experimental design**

This study aimed to examine how the informativeness of another's non-communicative eye movements influences joint attention using a gaze-contingent paradigm. There were two independent factors, the factor of context (Predictive, No-Search, Random) and stimulus (Eyes, Arrows), with a within-subjects design. The authors adapted the stimuli and task originally implemented by Caruana et al. This gaze-contingent paradigm of implicit joint attention would be useful to investigate spontaneous joint attention.

**Validity of the findings**

Line 173～   Participants: Have you conducted sample size estimation?

We based our sample size on multiple previous studies using similar paradigms across samples of 25-30 participants. In these studies, we reported large effect sizes of context ($\beta$=2.017, $\beta$=0.890). It is worth noting however, that these previous studies used a between-subjects design whereas our current study employed a more powerful, fully within-subjects design. As such, the target sample size of 30, which is at the higher end of our previous studies' sample sizes, was expected to provide a conservative estimate given that the current study employed a more powerful within-subjects design.

We have now added the following justification to lines (182-183):

> *"In line with previous studies using variations of the same task (Caruana et al., 2020; Caruana et al., 2017a), thirty-one adult participants were recruited from a pool of undergraduate psychology students at Macquarie University."*

What was the gender ratio of the participants? It should be clearly reported since some studies have reported gender differences in gaze sensitivity.

Thank you for your comment. Under 'Participants' line (179) we detail the following:

*"The final sample included 29 participants (M$_{age}$= 19.76 years; SD= 4.4; 8 males)."*

We have now added the following to make this information more explicit in line (190):

*"The final sample included 29 participants (21 females: 8 males; M$_{age}$= 19.76 years; SD= 4.4)."*

Line 277～

The authors mentioned that 6 people reported noticing the difference between Alan's and Tony's search patterns. Have you conducted analyses excluding these 6 participants?

Thank you for your question. Indeed, they were not excluded from the final analysis as we were interested in the effect of context regardless of whether participants consciously noticed the difference or not. We would also like to highlight that our LME model included random slopes and intercepts by subjects, which should account for any random sources of variability between participants. Our data is also available and these participants are clearly labelled in our dataset. We have now added a line (302-304) to clarify and it reads as follows:

*"Six people reported noticing the difference between Alan and Tony's search patterns. Their data was not excluded as our goal was to determine whether context effects manifested irrespective of whether participants consciously detected the variation in the search pattern or not. For a full summary of the subjective data, see documentation on our OSF Project page https://osf.io/e7kg8/."*

Line 310～ Statistical analyses

How many trials were excluded from the analyses?

We have now made this information explicitly clear in our revised manuscript in the section **Statistical Analyses** lines (339-347):

*"For accuracy analyses, 'Calibration' and 'Search' errors were removed (128 trials total) before analysing the remaining trials for the proportion of correct trials. This was done because these errors occurred before the relevant gaze or arrow cue was presented and, hence, do not represent true joint attention errors. The final accuracy analysis included 5092 trials in total. For SRT analyses we excluded incorrect trials (491 trials total) and trials where participants responded faster than 150 ms (464 trials total), as these were likely to be anticipatory responses (Carpenter, 1988). This resulted in the removal of 955 trials with 4265 trials included in the final SRT analyses. Raw eye-tracking data was screened and analysed in R using a custom script (https://osf.io/e7kg8/).*

Line 414～ Model fit analyses

Can you present data on model fitting such as the AIC and BIC? To compare the model fitting levels, each model's index of fitting should be reported.

We have edited the **Model fit analyses** section, lines (444-458) to include AIC and BIC parameters for each model, as requested:

> *"**Model fit analyses.** For quantifying the effects of stimulus and context, model-fit-improvement was compared as a function of each fixed effect parameter. Compared to the null model (i.e., a model with no fixed-effect factors; AIC = 11647.49, BIC = 11730.14), adding the context factor significantly improved the model fit by 36.05 times (AIC = 11615.43, BIC = 11710.81, $X^2(1) = 36.05$, p < 0.001). Adding the stimulus factor to the context-only model improved the model fit another 5.50 times (AIC = 11611.94, BIC = 11713.67, $X^2(1) = 5.50$, p = 0.019). On the other hand, including the stimulus factor to the null model first enhanced the model's fit by only 5.71 times (AIC = 11643.78, BIC = 11732.79, $X^2(1) = 5.71$, p = .017), while adding the context effect to the stimulus-only model significantly improved the model fit 35.84 times (AIC = 11611.94, BIC = 11713.67, $X^2(1) = 35.84$, p < .001). Critically, compared to a model containing fixed-effect factors for both stimulus and context, adding the interaction parameter significantly improved the model fit by 20.09 times (AIC = 11595.85, BIC = 11710.30, $X^2(1) = 20.09$, p < .001). These analyses show a markedly larger effect of context than stimulus. However, it also suggests that both factors explain unique variance in the data and that the data are significantly better explained by a model that specifies a stimulus-by-context interaction."*

Lines 485-489

The authors mentioned that experience may modulate responses to gaze cues. However, participants were faster to respond to Arrows than Eyes in general (Lines 396-397). It may be hypothesised that arrow cues cannot induce rapid responses if the amount of the experiences influences saccades. The authors should mention the different mechanisms of attention orientation between arrow- and gaze-cueing situations rather than discussing how experience modulates attentional responses

Thank you for your comment. First, we would like to clarify that our argument is not that people have more experience perceiving and responding to eye cues than arrow cues in general – although that might be the case. Rather, we entertain and present a more specific, possible explanation for the observed interaction effect. Our proposal is that we have more experience responding to eye gaze cues than arrows *in dynamic* interactive contexts. That is, where we respond to an informative and communicative cue that follows a sequence of other non-communicative cues, where we have to disregard irrelevant from relevant gaze shifts.

We have lots of experience with arrows providing relevant information (e.g., on signs), but very limited experience encountering *irrelevant* arrow cues, and arrow cues in *dynamic* sequences. You could even say, that we have more experience in real-word settings responding to singular arrow cues

than singular, non-dynamic gaze cues. This may, in part, explain why we see a basic advantage for arrows in our NoSearch task context, as you point out, where spatial cues are presented in isolation.

On the contrary, much of the spatial information conveyed by gaze shifts are uninformative from a communicative standpoint because people move their eyes constantly. Therefore, our claim here is simply that we may find it harder to disregard irrelevant arrow cues in dynamic contexts – and easier to disregard irrelevant gaze cues in dynamic contexts. Our data aligns with this claim. We also want to emphasise that our claim about experience is made to explain the interaction effect observed, in which we see a relative advantage for eyes in the dynamic contexts relative to the NoSearch baseline condition – where we see an inversion of this advantage effect (faster responses to arrows).

We have now revised this part of the **Discussion,** lines (500-504), to contextualise our hypothesis concerning experience more clearly. We firmly believe this possible explanation has merit and is worth mentioning so as to guide future research.

> *"**Context by Stimulus Interactions.**
>
> Although we found faster responses to Arrows than Eyes when no non-communicative eye movements were present (i.e., NoSearch context, discussed further below), we critically observed that a relative gaze advantage emerged when participants were required to respond to Eyes in dynamic contexts (i.e., Random and Predictive contexts) relative to the NoSearch baseline condition. Specifically, when comparing Random and NoSearch conditions, we observed slower response times in Random trials when responding to Arrows but not Eyes. Further, when comparing Predictive and NoSearch trials, we observed relatively faster responses for Eyes than Arrows. Together, these findings suggest that people are better positioned to adaptively use non-communicative gaze information when it offers opportunities to facilitate joint attention, and disregard this information when it does not, for Eyes but not Arrows in otherwise analogous task contexts.*

We have also made the following revision to further include the possible role of experience in explaining the faster reaction time to Arrows than Eyes in the NoSearch context, lines (575-583):

> *"It is noteworthy, however, that despite implementing more perceptually matched stimuli in the current study, we still find evidence for significantly faster reaction times to Arrows than Eyes in the NoSearch task, consistent with our previous findings. Previously, we discussed the possible role of experience in parsing and responding to Eye and Arrow spatial cues in dynamic contexts (i.e., Random and Predictive contexts). In the same way that we may have more experience responding to relevant eye cues than arrows in dynamic contexts, it is also likely the case that most people have more experience observing and responding to static arrows than eyes in real-world contexts. This might explain why we see faster responses to Arrows than Eyes in the NoSearch task where non-dynamic spatial cues were presented to participants. "*

We also acknowledge that there might be different reasons for why people might respond differently to eyes and arrows in non-interactive contexts. We agree that there might be potential bottom-up mechanisms that influence this behaviour, and we note this in our discussion when we later discuss the basic arrow advantage effect you mention – and that is observed on NoSearch trials, lines (583-591).

> *"Alternatively, it is also possible that slower responses to eyes in the NoSearch condition are due to eye contact's cascading effect on downstream attention and social cognition processing (Dalmaso et al., 2017; Gale et al., 1975; Helminen et al., 2011; Hietanen et al., 2020; Kleinke & Pohlen, 1971; Nichols & Champness, 1971). This may increase cognitive processing load by automatically engaging higher-order social-cognitive processing (Burra et al., 2019; Conty et al., 2007; Kampe et al., 2003; Mares et al., 2016; Schilbach et al., 2006; Senju & Johnson, 2009). As such, the presence of direct gaze during non-dynamic tasks may contribute to slower responses to Eyes than Arrows due to slower disengagement from direct gaze stimuli."*

However, we are not convinced that engaging in further discussion of the gaze-cueing literature would be a constructive endeavour in the context of this paper for two reasons. First, our study examines intentional responses to communicative gaze cues during joint attention interactions. As we have discussed at length in our previous papers, gaze-cuing studies probe very different attention mechanisms than those engaged during joint attention; specifically, they measure the extent to which spatial cues (eyes or arrows) result in reflexive shifts in attention (Caruana et al., 2020; Caruana et al., 2017b), whereas joint attention involves the intentional response to communicative gaze signals. Second, the gaze-cuing literature has failed to reveal any systematic formulation for basic differences in attention processes engaged when people are cued using eye or arrow stimuli (see Chacón-Candia et al., 2022 for a recent meta-analysis). We now emphasise this point in the discussion, lines (512-516).

> *"Existing evidence from studies using traditional gaze-cueing paradigms that investigate more reflexive gaze orientation have failed to identify reliable differences in responses between gaze and arrow cue conditions (see meta-analysis for a review, Chacón-Candia et al., 2022). Our study, however, presents evidence for a specific advantage for gaze when joint attention bids were embedded in a more naturalistic and dynamic stream of information. There are a few possible explanations for this gaze advantage."*

Lines 528-530

Although the authors mentioned the limitation that the eye contact stimulus was more visually salient than the fixation point, in that case, other results of the difference between Arrow and Eyes also have to be more moderate. Again, I recommend discussing the difference in attention orientation mechanisms between Arrow and Gaze cues.

Thank you for your comment. We believe that this has now been addressed in our response to the previous comment.

**Additional comments**
Thank you for providing me the opportunity to review this interesting study. Especially, the gaze-contingent paradigm used in this study has potential interest for researchers in social cognition. Since this task can measure spontaneous joint attention, it would be interesting to use the same task in people with ASD.

Thank you – and we agree. We highlight this in the future implications section (see next response) and indeed this work is currently underway in our lab.

As mentioned above, I just have some concerns about the statistics and discussion. It would be challenging to integrate the current results, however, I would like to see the authors' original theoretical contributions. The current discussion seems to explain each result separately rather than emphasising the theoretical contributions of this study.

We have now highlighted and cohesively summarised the theoretical implications of this study and explained how our findings may be used to further advance our theoretical conceptualisation of joint attention in lines (605 – 628) under **Implications and Future Applications**:

> "*Implications and Future Application. The current findings, and our updated paradigm, offers a new tool for carefully examining social information processing in dynamic contexts. Our findings provide a valuable starting point for constructing a relevance model of gaze-based joint attention that critically considers the dynamic contextual properties of gaze that shape social perception and action. This is of key theoretical importance given that our understanding of gaze perception to date has been built on findings from experimental work using non-interactive paradigms, in which gaze stimuli are presented in non-dynamic contexts. A new relevance model of gaze-based joint attention that accounts for the dynamic properties of gaze – and paradigms that can test these properties systematically – are critical for advancing our understanding of gaze use during realistic social interactions with artificial agents and between neurodiverse humans.*
>
> *For instance, this model can be instrumental in guiding research examining the factors that influence how humans perceive and respond to gaze information during interactions with artificial agents (e.g., virtual characters and social robots). Such research is critical for taking a human-centered approach to the design of artificial agents that can achieve intuitive interactions with humans. Critical to this future empirical endeavor is understanding the extent to which social cognitive processes (e.g., mental state attribution) have a top-down influence on the processing of communicative gaze signals during dynamic interactions.*

*Future studies could begin to explore this using our paradigm by manipulating whether participants believe the avatar or arrow stimuli are being controlled by a human partner or computer. This would help elucidate, for instance, the extent to which mental state attribution underpins the relative advantage for eye gaze observed in dynamic contexts.*

*In moving towards more ecologically-valid conceptualizations of joint attention, our approach also offers opportunities for studying joint attention difficulties experienced by autistic people (e.g. Charman, 2003; Hobson & Hobson, 2007; Mundy et al., 1994), who often report marked subjective difficulty in establishing eye contact with others (Trevisan et al., 2018; also Adrien et al., 1993; Kanner, 1943; Mirenda et al., 1983; Zwaigenbaum et al., 2005). Specifically, our paradigm can be used to interrogate why – and under what conditions – autistic individuals may experience difficulties responding to gaze-cued joint attention bids. Early studies have suggested this may be due to a difficulty in differentiating communicative and non-communicative gaze information (Caruana et al., 2018). The current paradigm would enable this to be directly interrogated to better understand the unique challenges that some autistic individuals face in processing social cues."*

**Reviewer 3 (Anonymous)**

**Basic reporting**

This paper illustrates an eye-tracking study in which participants were engaged in a gaze-contingency task with an avatar (social condition) or a dynamic arrow (non-social condition). The main results show that the participants were influenced by the gaze of the avatar when looking for a target, and this was particularly evident when the gaze of the avatar was informative about the location of the target. In addition, these results were much more evident in the social than in the non-social condition.

I have appreciated this paper, which is well written and organized. I only proposed some comments aimed at increasing some theoretical motivations underlying this work.

Thank you.

**1) Introduction**

- Line 50-51: there are a couple of more recent and relevant reviews on this topic that I'm missing (Dalmaso, Castelli, et al., 2020; McKay et al., 2021)

Thank you for pointing this out. We have now included these references in the **Introduction** lines (51 – 52):

> *"Until recently, most experimental studies of 'joint attention' required participants to observe and respond to a single gaze cue on each trial, often in a non-interactive context (see Caruana et al., 2017b; Frischen et al., 2007; Nation & Penny, 2008; Dalmaso, Castelli, & Galfano, 2020; McKay et al., 2021 for reviews)."*

I have appreciated the fact that you also introduced the control non-social conditions with arrows, which is important to establish whether a given phenomenon is truly social or not; however, I feel that the theoretical motivation for this control condition should be better motivated; for instance, there is a recent meta-analyses showing that gaze and arrow seems to lead to vert similar results in social attention tasks, at least at the behavioral level (Chacón-Candia et al., 2022)

Thank you for your comment. We agree that the rationale could be made clearer. In sum, previous gaze-cueing studies have shown no difference between eyes and arrows, but no previous studies have investigated this effect in a dynamic and interactive context where participants are required to evaluate a dynamic stream of relevant and irrelevant spatial information. We have tried to articulate this more clearly in the Introduction (lines 124 – 132) so that it reads as follows:

*"Participants also completed non-social versions of these tasks in which gaze cues were replaced with arrow cues. Even though previous studies indicate no reliable differences in response times in cueing studies comparing eye gaze and arrow cue conditions, it is important to note that these studies used non-interactive gaze-cueing paradigms (see meta-analysis for a review, Chacón-Candia et al., 2022). It was, hence, important to investigate if we find different effects in dynamic contexts that reflect the contexts in which we typically observe and respond to gaze information during dyadic, face-to-face interactions (i.e., where the eye gaze of others is constantly changing in both informative and uninformative ways), and where the participants goal is to intentionally respond to this spatial information."*

Related to my previous comment, I'm wondering why in the control 'non-social' condition you decided to maintain the picture of the face with closed eyes; I can understand the advantage of presenting participants with a similar perceptual stimulus, but at the same time I'm wondering if this condition can be actually labelled as truly 'non-social'

We appreciate your comment and acknowledge your concern. Indeed, this was an aspect of our experimental design we deliberated on at length before conducting the experiment. Ultimately, our decision to include this visual element was made to be consistent with the approach taken in previous studies, aimed at maintaining visual complexity across conditions. We acknowledge the possible influence of the face image on the perceived 'socialness' of the arrows. However, we don't believe this has any meaningful impact on the validity of our findings, nor the conclusions we draw. This is because, if the presence of the face were to make the arrows appear *more social*, it would theoretically lead to smaller differences in participants' responses across the Eye and Arrow conditions. However, our study consistently demonstrates robust relative differences between these conditions, suggesting that any potential influence of the face image did not override the primary effects of interest. That is, removing the face stimuli on Arrow trials, if anything, would likely make the observed effects larger. In this way, we view our study's design as a more conservative and controlled approach, which allowed us to focus on the specific manipulation of non-communicative gaze cues while maintaining consistency with prior research.

**Methods**

Line 201-202: you primed participants with a cover story saying that it was an interactive game with another person placed in another laboratory; I'm wondering if this was necessary to observe the results you reported here; do you think that a different pattern could emerge if no cover story was provided? This manipulation reminds me of some work showing that 'mental state' attribution can shape responses to social attention responses (e.g. Wiese et al., 2014).

Thank you for this comment. Indeed, we are aware of evidence reporting differences in responses based on intentional stance and mental state attribution using both variations of our own paradigm as well as gaze-cueing paradigms. We have now added the following to the **Design and procedure** section lines (213 – 218) to clarify and support:

> *"Participants were told that they would be playing a collaborative game with two different members of the research team, named 'Alan' and 'Tony', who would be interacting with them from an adjacent laboratory. This cover story was important since previous studies using both more traditional gaze-cueing paradigms and variations of our own paradigm have shown differences in subjective experience (Caruana, Spirou, et al., 2017), behavioural responses (Caruana, Spirou, et al., 2017; Morgan et al., 2018; Teufel et al., 2010; Wiese et al., 2014), and neurological responses (Caruana & McArthur, 2019; Caruana, de Lissa, 2017) when participants believe their virtual partner is controlled by a human rather than a computer.*

We have also added a new section to our Implications section at the end of the Discussion to highlight the potential value in manipulating participants' beliefs in the specific context of the current study's paradigm to determine whether the observed context interactions are influenced in part by the engagement of mentalising processes. Lines (620-626).

> *"Critical to this future empirical endeavor is understanding the extent to which social cognitive processes (e.g., mental state attribution) have a top-down influence on the processing of communicative gaze signals during dynamic interactions. Future studies could begin to explore this using our paradigm by manipulating whether participants believe the avatar or arrow stimuli are being controlled by a human partner or computer. This would help elucidate, for instance, the extent to which mental state attribution underpins the relative advantage for eye gaze observed in dynamic contexts."*

Please clarify how the sample size was established (e.g., power analysis?); I am wondering if N = 31 (which seems relatively small) is enough to detect the effects that you wanted to examine.

We based our sample size on multiple previous studies using similar paradigms across samples of 25-30 participants. In these studies, we reported large effect sizes of context ($\beta$=2.017, $\beta$=0.890). It is worth noting however, that these previous studies used a between-subjects design whereas our current study employed a more powerful, fully within-subjects design. As such, the target sample size of 30, which is at the higher end of our previous studies' sample sizes, was expected to provide a conservative estimate given that the current study employed a more powerful within-subjects design.

We have now added the following justification to lines (182-183):

*"In line with previous studies using variations of the same task (Caruana et al., 2020; Caruana et al., 2017a), thirty-one adult participants were recruited from a pool of undergraduate psychology students at Macquarie University."*

RTs analyses: you discarded trials with an RT < 150 ms (i.e., anticipations); what about trials with an excessive latency (i.e., outliers? If any). Please clarify.

Thank you for this comment. We closely examined the data and identified that all the remaining responses were valid and representative of participants' actual response profiles. We did not find the need to define any threshold for excessive latency especially since this was a within-subjects design. We added the following to **Statistical Analyses** section lines (339 – 347) to clarify:

*"For accuracy analyses, 'Calibration' and 'Search' errors were removed (128 trials total) before analysing the remaining trials for the proportion of correct trials. This was done because these errors occurred before the relevant gaze or arrow cue was presented and, hence, do not represent true joint attention errors. This resulted in analysing accuracy on 5092 trials in total. For SRT analyses we excluded incorrect trials (491 trials total); and trials where participants responded faster than 150 ms (464 trials total), as these were likely to be anticipatory responses (Carpenter, 1988). No other outliers were identified. This resulted in the removal of 955 trials with 4265 trials included in the SRT analyses. Raw eye-tracking data was screened and analysed in R using a custom script (https://osf.io/e7kg8/)."*

**Results**

As 6 participants reported differences between 'Alan' and 'Tony', I am wondering if these people have been included in the final analyses.

Thank you for pointing this out. Indeed, they were included in the final analysis as we were interested in the effect of context regardless of whether they consciously noticed the difference or not. We would also like to highlight that our LME model included random slopes and intercepts by subjects, which should account for any variations between participants. Also note that all our data is made available and these participants are clearly noted. We have now added a line (302-304) to clarify and it reads as follows:

*"Indeed, 6 people reported noticing the difference between Alan and Tony's search patterns. Their data was not excluded as our goal was to determine whether context effects manifested irrespective of whether participants consciously detected the variation in the search pattern or not. For a full summary of the subjective data, see documentation on our OSF Project page https://osf.io/e7kg8/."*

I think it would be better to report p < .001 instead of p <.000

Thank you for your comment. We have made the suggested changes.

Line 397: technically speaking, p = .05 is a non-significant result; please verify.

This is a valid comment. However, given the arbitrary nature of a <.05 threshold, we believe that this effect is noteworthy (see Cohen, 1994; Lakens & Cladwell, 2021 for a relevant discussion). We have revised the manuscript nonetheless to use more tentative language (425 – 428):

> *"We also found that, overall, participants were faster when responding to Arrows than Eyes, however this effect was marginally significant, since it was at our defined threshold for statistical significance ($\beta$ = -0.16, t = -2.02, p =.050)."*

**Discussion**

Line 508-509: I am missing another relevant work showing that eye contact can also increase the oculomotor response in a gaze cueing task (Dalmaso, Alessi et al., 2020)

Thank you for pointing this out. We have now revised this part of the manuscript to include this (546 - 547) in the **Discussion**:

> *"This explanation also aligns with past work using gaze cueing paradigms which have shown faster oculomotor response after the observation of eye contact (Dalmaso, Alessi et al., 2020; Xu, Zhang, & Geng, 2018; Kompatsiari, Ciardo, Tikhanoff, Metta, & Wykowska, 2018)."*

Lines 542-546: Please note that there is indeed evidence for the notion that direct-gaze stimuli can hold attention as reflected in oculomotor indexes; please see (Dalmaso et al., 2017) which could be briefly mentioned.

Thank you for pointing this out. We have now included this reference to line (585) in the **Discussion**:

> *"It is noteworthy, however, that despite implementing more perceptually matched stimuli in the current study, we still find evidence for significantly slower reaction times to Eyes than Arrows in the NoSearch task, consistent with our previous findings. As previously discussed, eye contact has a cascading effect on downstream attention and social cognition processing (Dalmaso et al., 2017; Gale et al., 1975; Helminen et al., 2011; Hietanen et al., 2020; Kleinke & Pohlen, 1971; Nichols & Champness, 1971)."*

References

Chacón-Candia, J. A., Román-Caballero, R., Aranda-Martín, B., Casagrande, M., Lupiáñez, J., & Marotta, A. (2022). Are there quantitative differences between eye-gaze and arrow cues? A meta-analytic answer to the debate and a call for qualitative differences. Neuroscience & Biobehavioral Reviews, 104993. https://doi.org/10.1016/j.neubiorev.2022.104993

Dalmaso, M., Alessi, G., Castelli, L., & Galfano, G. (2020). Eye contact boosts the reflexive component of overt gaze following. Scientific Reports, 10(1), 4777. https://doi.org/10.1038/s41598-020-61619-6

Dalmaso, M., Castelli, L., & Galfano, G. (2017). Attention holding elicited by direct-gaze faces is reflected in saccadic peak velocity. Experimental Brain Research, 235(11), 3319–3332. https://doi.org/10.1007/s00221-017-5059-4

Dalmaso, M., Castelli, L., & Galfano, G. (2020). Social modulators of gaze-mediated orienting of attention: A review. Psychonomic Bulletin & Review, 27(5), 833–855. https://doi.org/10.3758/s13423-020-01730-x

McKay, K. T., Grainger, S. A., Coundouris, S. P., Skorich, D. P., Phillips, L. H., & Henry, J. D. (2021). Visual attentional orienting by eye gaze: A meta-analytic review of the gaze-cueing effect. Psychological Bulletin, 147(12), 1269–1289. https://doi.org/10.1037/bul0000353

Wiese, E., Wykowska, A., & Müller, H. J. (2014). What we observe is biased by what other people tell us: beliefs about the reliability of gaze behavior modulate attentional orienting to gaze cues. PloS One, 9(4), e94529. https://doi.org/10.1371/journal.pone.0094529

Experimental design
Please see my previous comments
Validity of the findings
Please see my previous comments